# A First-Principles Investigation on the Structural, Optoelectronic, and Thermoelectric Properties of Pyrochlore Oxides (La_2_Tm_2_O_7_ (Tm = Hf, Zr)) for Energy Applications

**DOI:** 10.3390/ijms232315266

**Published:** 2022-12-03

**Authors:** Zeesham Abbas, Sajjad Hussain, Shabbir Muhammad, Saifeldin M. Siddeeg, Jongwan Jung

**Affiliations:** 1Department of Nanotechnology and Advanced Materials Engineering, Sejong University, Seoul 05006, Republic of Korea; 2Department of Chemistry, College of Science, King Khalid University, P.O. Box 9004, Abha 61413, Saudi Arabia

**Keywords:** pyrochlore oxides, DFT + U, first-principles investigations, photovoltaic, thermoelectric properties

## Abstract

A first-principles calculation based on DFT investigations on the structural, optoelectronic, and thermoelectric characteristics of the newly designed pyrochlore oxides La_2_Tm_2_O_7_ (Tm = Hf, Zr) is presented in this study. The main quest of the researchers working in the field of renewable energy is to manufacture suitable materials for commercial applications such as thermoelectric and optoelectronic devices. From the calculated structural properties, it is evident that La_2_Hf_2_O_7_ is more stable compared to La_2_Zr_2_O_7_. La_2_Hf_2_O_7_ and La_2_Zr_2_O_7_ are direct bandgap materials having energy bandgaps of 4.45 and 4.40 eV, respectively. No evidence regarding magnetic moment is obtained from the spectra of TDOS, as a similar overall profile for both spin channels can be noted. In the spectra of $\varepsilon_{2}\left(\omega \right)$, it is evident that these materials absorb maximum photons in the UV region and are potential candidates for photovoltaic device applications. La_2_Tm_2_O_7_ (Tm = Hf, Zr) are also promising candidates for thermoelectric device applications, as these p-type materials possess ZT values of approximately 1, which is the primary criterion for efficient thermoelectric materials.

## 1. Introduction

Pyrochlore oxides with the formula A_2_B_2_O_7_ (A_2_B_2_O_6_O’) have emerged as potential multiferroic materials due to the tunability of their structures and local displacement of their B-site atoms inside the original structure of these materials [1]. The stability of the pyrochlore oxides depends on the size ratio of the two cations occupying the A and B sites [2]. The ideal pyrochlore structures have eight molecules per unit cell and belong to the Fd-3m space group. In the pyrochlore structures, pentavalent/tetravalent cations can occupy the B-site; divalent/trivalent cations can occupy the A-site [3]. Every atom in a crystalline structure occupies a distinct position known as the Wyckoff position [4]; therefore, A-site, B-site, O’- and O-atoms are located at the *16c*, *16d*, *48f* and *8a* sites, respectively [3]. However, the 8b-site remains vacant. Due to their remarkable ferroelectric properties, many pyrochlore materials, including Nd_2_Zr_2_O_7_, Bi_2_T_i2_O_7_, Cd_2_Re_2_O_7_, and La_2_Zr_2_O_7_, have been reported in the literature [5,6]. Ekaterina A. et al. [7] presented a series of newly synthesized pyrochlore oxides La_2-x_Sr_x_Zr_2_O_7-δ_ (x = 0, 0.05, 0.1, 0.15 and 0.5) by the citrate–nitrate procedure. Amalesh K. et al. [8] reported the effect of variable doping concentrations of Pr^3+^ ions on the photoluminescence properties of La_2_Zr_2_O_7_:Pr^3+^ phosphor materials. Pyrochlore oxides have been employed for various technological applications such as anti-erosion of Ag [9], magnetic devices [10], photoluminescence devices [11], thermal barrier coatings of diesel engines [12], gas sensors [13], high-temperature catalysts [14], enhanced photocatalytic applications [15] and nuclear waste disposal [16], due to their significant structural flexibility.

The quest for academia and the industry is to find renewable and novel energy sources, as there is a never-ending hike in global energy demands. Photovoltaic materials are promising candidates for applications in clean and renewable energy technologies. Kisa F. et al. [4] presented a comprehensive DFT study on optoelectronic properties of pyrochlore oxides La_2_Q_2_O_7_ (Q = Ge, Sn). They reported that these materials are active optical materials and are promising candidates for photovoltaic devices working in UV or visible regions. Thermoelectric materials can be employed in diverse applications to address the international energy crisis, as these materials use excess heat energy for clean and sustainable energy generation [17]. The main quest of the researchers working in the field of renewable energy is to manufacture suitable materials for commercial applications such as thermoelectric and optoelectronic devices. These materials must possess excellent physical and chemical properties. Re-X-O ternary systems exhibit numerous desirable properties for electrochemistry and catalysis, where Re and X represent rare-earth elements and metals from the platinum group [18]. Many ternary Re-X-O materials have widely been studied at high temperatures for their thermodynamic properties and phase equilibrium [19]. La_2_Pd_2_O_7_, a member of this ternary group, is a promising chemical with numerous potential applications [17]. Ca_1̶x_Bi_x_Pd_3_O_4_ and Ca_1̶x_Li_x_Pd_3_O_4_, n- and p-type semiconductors, respectively, are fascinating prospects for thermoelectric applications. Due to their large effective mass, these materials’ primary drawback for thermoelectric (TE) applications is their low mobility [20]. Powder diffraction, in combination with a simple modeling technique, was used by Attfield et al. to examine crystalline structures of lanthanum palladium oxides such as La_4_PdO_7_ and La_2_Pd_2_O_5_ [21]. Guilin W. et al. [7] reported a thorough study on the thermophysical and mechanical properties of La_2_Zr_2_O_7_.

Pyrochlore oxides have appropriate characteristics for TE device applications; in addition, they are efficient absorbers of incident photons. This work used first-principles-based DFT calculations to study the optical, electronic, and thermoelectric properties of pyrochlore oxides La_2_Tm_2_O_7_ (Tm = Hf, Zr). To gain an insight into the future applications of pyrochlore oxides in thermoelectric devices and solar cells, their ground state properties were studied using the FP-LAPW technique. The fundamental motivation behind this study is the growing interest of scientists in pyrochlore oxides as potential materials for renewable energy applications. This family of complex materials is fascinating for photovoltaic and thermoelectric device applications due to their energy conversion efficiency depending on their high absorption capability and figure of merit. They are promising candidates for UV photovoltaic devices due to the relatively wide band gaps of La_2_Tm_2_O_7_ (Tm = Hf, Zr) and their structural stability. The following article is designed in such a way that a discussion of the results is provided in Section 2, computational details in Section 3, and a summary of the discussed results are provided in Section 4 (conclusions).

## 2. Results and Discussion

The calculated results for the optoelectronic, thermoelectric and structural properties of La_2_Tm_2_O_7_ (Tm = Hf, Zr) are presented and discussed in this manuscript section. We explored the capability of these pyrochlore oxides for optoelectronic and thermoelectric device applications to produce clean energy. The structural stability of La_2_Tm_2_O_7_ (Tm = Hf, Zr) is determined using the calculated structural properties of these compounds. The potential of the pyrochlore oxides for optoelectronic applications such as solar cells can be investigated using calculated optoelectronic properties. Furthermore, the usability of these materials in thermoelectric devices such as thermal generators can be determined using their thermoelectric properties. Based on the presented results, these pyrochlore oxides are promising materials for thermoelectric and optoelectronic device applications.

### 2.1. Structural Properties

In this manuscript section, we discuss essential parameters to check the structural stability of La_2_Tm_2_O_7_ (Tm = Hf, Zr) using the calculated structural properties. The structural optimization parameters for La_2_Tm_2_O_7_ (Tm = Hf, Zr) are calculated using the Birch-Murnaghan equation of state (EOS) presented in Equation (1) [22]. The optimized volume versus energy curves for the unit cells of pyrochlore oxides is presented in Figure 1. The ground state energy of the material corresponds to the lowest point in the volume–energy spectra, and the volume at that point is optimized. The lattice parameters corresponding to the equilibrium volume are known as optimized lattice parameters. Table 1 and Table 2 present geometric and optimized parameters calculated for pyrochlore oxides La_2_Tm_2_O_7_ (Tm = Hf, Zr).
(1)$$E_{tot}\left(V\right)=E_{0}\left(V\right)+\frac{B_{0}V}{B^{\prime }\left(B^{\prime }-1\right)}\left[B\left(1-\frac{V_{0}}{V}\right)+{\left(\frac{V_{0}}{V}\right)}^{B^{\prime }}-1\right]$$

### 2.2. Electronic Properties

The calculated electronic properties, i.e., energy band structures and the density of states (DOS) for La_2_Tm_2_O_7_ (Tm = Hf, Zr), are presented and discussed in this manuscript. The energy band structures of the compounds mentioned above are plotted along the high symmetric axis of the IBZ (irreducible Brillouin zone) on a continuous energy range of −3 to 5 eV. DOS spectra for both compounds are presented and discussed on a continuous energy range of −6 to 6 eV to obtain insight into the most feasible electronic transitions between CB and VB.

#### 2.2.1. Energy Band Structures

The valuable information regarding ground state properties of pyrochlore oxides La_2_Tm_2_O_7_ (Tm = Hf, Zr), such as optical and charge transport properties, can be obtained from the calculated energy band structures. A continuous energy range of −3 to 5 eV is used to plot the calculated energy band structures by taking the Fermi level ($E_{F}$) at 0 eV (presented in Figure 2). High symmetry points in IBZ are used to calculate energy band structures using GGA + U approximation in the DFT. In Figure 2a,b, the calculated energy band structures using GGA + U approximation for La_2_Hf_2_O_7_ and La_2_Zr_2_O_7_, respectively, are presented. This study shows that La_2_Tm_2_O_7_ (Tm = Hf, Zr) are wide-bandgap semiconductors. The calculated values of energy band gaps for La_2_Hf_2_O_7_ and La_2_Zr_2_O_7_ are approximately 4.45 and 4.40 eV, respectively. Both pyrochlore oxides La_2_Tm_2_O_7_ (Tm = Hf, Zr) are direct band semiconductors, as both VBM (valence band maxima) and CBM (conduction band minima) occur at the same symmetric point Γ. The potential of the materials for technical applications such as thermoelectric, magneto-electronic, and optoelectronic devices can be analyzed using in-depth information regarding the energy band structures and gaps.

#### 2.2.2. Density of States

Deep knowledge regarding the nature of energy states and the various atomic orbitals that are contributing to these states can be obtained from the calculated density of states (DOS) for that compound. Detailed analyses of TDOS (total density of states) and PDOS (partial density of states) are required to understand the composition of VB and CB. The spectra of TDOS for La_2_Tm_2_O_7_ (Tm = Hf, Zr) are presented in Figure 3. The energy range of −6.0 to 6.0 eV is used to plot the investigated results of TDOS and PDOS. From Figure 3, we can state that there is no noticeable magnetic moment in La_2_Tm_2_O_7_ (Tm = Hf, Zr), as the spectra of TDOS are identical in both spin channels over the entire energy range. Hence, only results for the spin-up channel are presented in this study. The Fermi level in all electronic results is set at 0 eV.

The spectra of PDOS for La_2_Hf_2_O_7_ are presented in Figure 4. The PDOS spectra can validate the results obtained from the energy band structure evaluation. In the valence band of La_2_Hf_2_O_7_, significant contributions from O-atoms are evident from the spectra of PDOS. The O [2p^4^] and Hf [5d^2^] orbitals contribute mainly to the valence band, whereas La [5p^6^], La [5d^1^], and Hf [5p^6^] orbitals show minor contributions. In the conduction band of La_2_Hf_2_O_7_, significant contributions from La-atoms are evident from the spectra of PDOS, with minor contributions from Hf-atoms. The La [5d^1^] and Hf [5d^2^] orbitals contribute mainly to the conduction band, whereas La [5p^6^] orbitals show minor contributions.

The spectra of PDOS for La_2_Zr_2_O_7_ are presented in Figure 5. In the valence band of La_2_Zr_2_O_7_, significant contributions from O-atoms are evident from the spectra of PDOS. The O [2p^4^] and Zr [4d^2^] orbitals contribute mainly to the valence band, whereas La [5p^6^], La [5d^1^], and Zr [4p^6^] orbitals show minor contributions. In the conduction band of La_2_Zr_2_O_7_, significant contributions from La-atoms are evident from the spectra of PDOS, with minor contributions from Hf-atoms. The La [5d^1^] and Zr [4d^2^] orbitals contribute mainly to the conduction band, whereas La [5p^6^] orbitals show minor contributions.

### 2.3. Optical Properties

The reaction of the crystalline materials while interacting with electromagnetic (EM) waves at different energies (E=hυ) can be explained using optical properties. Deep knowledge of the optical characteristics, such as the dispersion and absorption of the material, is necessary while designing photovoltaic devices such as solar cells, LEDs, optical fibers, etc. [22]. Dielectric function ε(ω) presented in Equation (2) is a vital parameter to model the impact of EM radiation on a crystalline material [23]. ε(ω) is the fundamental input parameter for computing the remaining optical characteristics. The dielectric function ε(ω) consists of two parts: $\varepsilon_{2}\left(\omega \right)$ (imaginary/absorptive) and $\varepsilon_{1}\left(\omega \right)$ (real/dispersive) part.
(2)$$\varepsilon \left(\omega \right)=\varepsilon_{1}\left(\omega \right)+i\varepsilon_{2}\;\left(\omega \right)$$

The material exhibits both interband and intraband electronic transitions. Interband transitions can be direct or indirect and typically occur in semiconductor materials, whereas intraband transitions are typically seen in metallic compounds. The phonon scattering effect is related to indirect transitions. The complex dielectric function ε(ω) can be determined using the calculated energy band structure for La_2_Tm_2_O_7_ (Tm = Hf, Zr). The imaginary part $\varepsilon_{2}\left(\omega \right)$ involves all probable band transitions (between populated valence states (VS) and vacant conduction states (CS)), and Equation (3) can be used to calculate $\varepsilon_{2}\left(\omega \right)$ [24].
(3)$$\varepsilon_{2}^{ij}\left(\omega \right)=\frac{4\pi^{2}e^{2}}{Vm^{2}\omega^{2}}\times \sum_{knn^{\prime }\sigma }\langle kn\sigma |p_{i}|kn^{\prime }\sigma \rangle \langle kn^{\prime }\sigma |p_{j}|kn\sigma \rangle \times f_{kn}\left(1-f_{kn^{\prime }}\right)\sigma \left(E_{kn^{\prime }}-E_{kn}-\hbar \omega \right)$$

The calculated values of $\varepsilon_{2}\left(\omega \right)$ can be used to calculate $\varepsilon_{1}\left(\omega \right)$ using the Kramers–Kronig relation shown in Equation (4) [25]. The $\varepsilon_{1}\left(\omega \right)$ and energy bandgap are in inverse relation with one another.
(4)$$\varepsilon_{1}\left(\omega \right)=1+\frac{2}{\pi }P\int_{0}^{\infty }\frac{\omega^{\prime }\varepsilon_{2}\left(\omega^{\prime }\right)}{{\omega^{\prime }}^{2}-\omega^{2}}d\omega^{\prime }$$

The spectra of $\varepsilon_{1}\left(\omega \right)$ can be used to explain a significant case of the dispersion of incident photons. The calculated spectra of $\varepsilon_{1}\left(\omega \right)$ for La_2_Tm_2_O_7_ (Tm = Hf, Zr) are presented in Figure 6a. An energy range of 0 to 14 eV is used to plot all the studied optical parameters. At zero energy (ω=0), we can observe definite values of $\varepsilon_{1}\left(\omega \right)$, known as zero frequency limit or static values of $\varepsilon_{1}\left(0\right)$. Penn’s model equation shown in Equation (5) can be used to establish a correlation between the energy bandgap and $\varepsilon_{1}\left(0\right)$.
(5)$$\varepsilon_{1}\left(0\right)=1+{\left(\frac{\hbar \omega_{p}}{E_{g}}\right)}^{2}$$

From Figure 6a, we can note that the static values of $\varepsilon_{1}\left(0\right)$ are 2.24 and 2.31 for La_2_Hf_2_O_7_ and La_2_Zr_2_O_7_, respectively. After that, peaks rise and reach the maximum value of around 5.0 eV. The spectra of $\varepsilon_{1}\left(\omega \right)$ then decrease sharply and enter the negative region at 11.82 and 11.5 eV for La_2_Hf_2_O_7_ and La_2_Zr_2_O_7_, respectively. The region where $\varepsilon_{1}\left(\omega \right)$ is negative can be used to characterize screened plasma, and plasmon plasma is the frequency where $\varepsilon_{1}\left(\omega \right)$ intersects the zero level (the dotted line). Materials show metallic and dielectric behavior below and above this dotted line [4].

The spectra of $\varepsilon_{2}\left(\omega \right)$ can be used to explain a significant case of the absorption of incident photons. The calculated spectra of $\varepsilon_{2}\left(\omega \right)$ for La_2_Tm_2_O_7_ (Tm = Hf, Zr) are presented in Figure 6b. In the spectra of $\varepsilon_{2}\left(\omega \right)$, it is evident that initially, values of $\varepsilon_{2}\left(\omega \right)$ are zero, and then peaks start to emerge from 3.28 and 3.17 eV for La_2_Hf_2_O_7_ and La_2_Zr_2_O_7_, respectively. The value from where the peak emerges is known as the optical bandgap of the material or threshold energy of $\varepsilon_{2}\left(\omega \right)$. Both energy bandgaps and threshold energies show good agreement with each other. From Figure 6b, we can note a sudden increase in the spectra of $\varepsilon_{2}\left(\omega \right)$ after threshold energies and move to the maximum values around 7.0 eV. Therefore, we can conclude that La_2_Tm_2_O_7_ (Tm = Hf, Zr) are efficient photon absorbers in the near UV region.

The calculated values of $\varepsilon_{2}\left(\omega \right)$ and $\varepsilon_{1}\left(\omega \right)$ can be employed to calculate other significant optical parameters such as the extinction coefficient K(ω) and refractive index n(ω) using Equations (6) and (7), respectively [26].
(6)K(ω)=12(−ε1(ω)+ε12(ω)+ε22(ω))12
(7)n(ω)=12(ε1(ω)+ε12(ω)+ε22(ω))12

The ratio of the speed of light in vacuum (c) to the speed of light in a medium (v) is known as the refractive index, n=c/v. This optical parameter is used to determine whether a material is acceptable for technological optical devices. Similar to $\varepsilon_{1}\left(\omega \right)$, deep knowledge regarding the dispersion of incident photons can also be obtained from the spectra of n(ω). The calculated spectra of n(ω) for La_2_Tm_2_O_7_ (Tm = Hf, Zr) are presented in Figure 6c. At zero energy (ω=0), we can observe definite values of n(ω), known as zero frequency limit or static values of n(0). Penn’s model equation shown in Equation (5) can be used to establish a correlation between the energy bandgap and $\varepsilon_{1}\left(0\right)$. Equation (8) can verify static values of n(0).
(8)$$n\left(0\right)=\sqrt{\varepsilon_{1}\left(0\right)}$$

From Figure 6a, we can note that the static values of n(0) are 1.5 and 1.52 for La_2_Hf_2_O_7_ and La_2_Zr_2_O_7_, respectively. The material is considered to be an active optical material when the value of its refractive index is between 1.0 and 2.0. After that, peaks rise and reach the maximum values of around 5.0 eV. The *n(ω)* spectra then decrease sharply and become less than unity at 10.19 and 11.1 eV for La_2_Hf_2_O_7_ and La_2_Zr_2_O_7_, respectively. In the region where n(ω) is less than unity, materials show metallic behavior. When (n<1), these materials exhibit the superluminal phenomenon that can be theoretically and experimentally observed. This unnatural phenomenon communicates that the speed of light in a vacuum is less than that of light in the medium, $c<v_{g}$, which is impossible [27]. Material with high optical conductivity values, refractive index, and absorption coefficients with low emissivity can be utilized in photovoltaic device applications.

An optical parameter that determines how effectively a material can absorb incident photons/radiations at a specific frequency is the extinction coefficient K(ω). Similar to $\varepsilon_{2}\left(\omega \right)$, deep knowledge regarding the absorption of incident photons can also be obtained from the spectra of K(ω). The calculated spectra of K(ω) for La_2_Tm_2_O_7_ (Tm = Hf, Zr) are presented in Figure 6d. In the *K*(*ω*) spectra, it is evident that initially, *K*(*ω*) values are zero, and then, peaks start to emerge from 2.33 and 3.43 eV for La_2_Hf_2_O_7_ and La_2_Zr_2_O_7_, respectively. The value from where the peak emerges is the known threshold energy of K(ω). From Figure 6b, we can note a sudden increase in the *K(ω*) spectra after threshold energies and move to the maximum values in the UV region. Therefore, we can conclude that La_2_Tm_2_O_7_ (Tm = Hf, Zr) are efficient photon absorbers in the near UV region.

Remaining optical parameters such as energy loss function L(ω), reflectivity coefficient R(ω), optical conductivity σ(ω), and absorption coefficient I(ω) can also be calculated using values of $\varepsilon_{2}\left(\omega \right)$ and $\varepsilon_{1}\left(\omega \right)$ [27].
(9)$$L\left(\omega \right)=-ln\left(\frac{1}{\varepsilon }\right)=\frac{\varepsilon_{2}\left(\omega \right)}{\varepsilon_{1}^{2}\left(\omega \right)+\varepsilon_{2}^{2}\left(\omega \right)}$$
(10)$$R\left(\omega \right)={\left|\frac{\sqrt{\varepsilon \left(\omega \right)}-1}{\sqrt{\varepsilon \left(\omega \right)}+1}\right|}^{2}=\frac{{\left(1-n\right)}^{2}+k^{2}}{{\left(1+n\right)}^{2}+k^{2}}$$
(11)$$\sigma \left(\omega \right)=\frac{\omega }{4\pi }\varepsilon_{2}\left(\omega \right)$$
(12)$$I\left(\omega \right)=\sqrt{2\omega }{\left\{-\varepsilon_{1}\left(\omega \right)+\sqrt{\varepsilon_{1}^{2}\left(\omega \right)+\varepsilon_{2}^{2}\left(\omega \right)}\right\}}^{\frac{1}{2}}$$

Some electrons undergo inelastic scattering when a beam of electrons is incident on the material. The energy lost by these electrons in the material can be interpreted as energy loss function L(ω). It is evident from Equation (9) that there exists an inverse relation between $\varepsilon_{2}\left(\omega \right)$ and L(ω). The calculated spectra of L(ω) for La_2_Tm_2_O_7_ (Tm = Hf, Zr) are presented in Figure 7a. In the *L*(*ω*) spectra, it is evident that initially, values of L(ω) are zero, and then, peaks start to emerge from 3.92 and 4.11 eV for La_2_Hf_2_O_7_ and La_2_Zr_2_O_7_, respectively. From Figure 7a, we can note a gradual increase in the *L*(*ω*) spectra after threshold energies and move to the maximum values around 13.11 and 13.28 eV for La_2_Hf_2_O_7_ and La_2_Zr_2_O_7_, respectively. These energies are also known as points of plasmon resonance for the compounds mentioned above.

The reflected-to-incident photons ratio is known as the reflectivity coefficient R(ω). The calculated spectra of R(ω) for La_2_Tm_2_O_7_ (Tm = Hf, Zr) are presented in Figure 7b. At zero energy (ω=0), we can observe definite values of R(ω) known as zero frequency limit or static values of R(0). From Figure 7b, we can note that the static values of R(0) are 0.039 and 0.043 for La_2_Hf_2_O_7_ and La_2_Zr_2_O_7_, respectively. It is evident from the spectra of R(ω) that these materials reflect a negligible number of incident photons (∼20%) up to 12.0 eV. La_2_Tm_2_O_7_ (Tm = Hf, Zr) reflects a maximum number of incident photons (∼55%) above 13.0 eV. Based on *R*(*ω*) spectra, it can also be concluded that these materials are promising candidates for absorption-related applications.

The property of the material to conduct electricity when exposed to light is known as optical conductivity σ(ω). The calculated spectra of σ(ω) for La_2_Tm_2_O_7_ (Tm = Hf, Zr) are presented in Figure 7c. In the spectra of σ(ω), it is evident that initially, values of σ(ω) are zero, and then, peaks start to emerge from 3.62 and 3.75 eV for La_2_Hf_2_O_7_ and La_2_Zr_2_O_7_, respectively. From Figure 7c, we can note a sudden increase in the spectra of σ(ω) after threshold energies and move to the maximum values around 7.44 and 9.13 eV for La_2_Hf_2_O_7_ and La_2_Zr_2_O_7_, respectively. It is evident from the spectra of σ(ω) that these materials are excellent conductors around the energies mentioned above.

Information regarding the penetration length of the incident photon in the material can be explained using absorption coefficient I(ω) when the energies of incident photons are greater than the energy band gap of the material. The calculated spectra of I(ω) for La_2_Tm_2_O_7_ (Tm = Hf, Zr) are presented in Figure 7d. In the *I*(*ω*) spectra, it is evident that initially, values of I(ω) are zero, and then, peaks start to emerge from 4.18 and 4.25 eV for La_2_Hf_2_O_7_ and La_2_Zr_2_O_7_, respectively. The value from where the peak emerges is known as the threshold energy of I(ω). From Figure 7d, we can note a sudden increase in the *I*(*ω*) spectra after threshold energies and moving to the maximum values in the UV region. Therefore, we can conclude that La_2_Tm_2_O_7_ (Tm = Hf, Zr) are efficient photon absorbers in the near UV region and can be utilized in photovoltaic devices working in the UV region.

### 2.4. Thermoelectric Properties

In recent years, thermoelectric materials have attracted massive attention from scientists due to the global energy crisis. The phenomenon of directly converting temperature gradient into electrical energy is called the thermoelectric effect. The thermoelectric (TE) properties are also included in this study. The traditional Boltzmann kinetic transport theory and the rigid band approximation are employed to calculate thermoelectric properties of pyrochlore oxides La_2_Tm_2_O_7_ (Tm = Hf, Zr) [28,29]. The TE behavior of pyrochlore oxides La_2_Tm_2_O_7_ (Tm = Hf, Zr) can be characterized using basic TE parameters such as the figure of merit $\left(ZT=\frac{S^{2}T}{\rho \kappa }\right)$, power factor ($PF=\sigma S^{2}$), Seebeck coefficient $\left(S=\frac{\Delta V}{\Delta T}\right)$, electronic thermal conductivity $\left(q=-k\frac{dT}{dx}\right)$ and electrical conductivity. The semi-classical theory-based Boltztrap code is employed to calculate aforesaid temperature-dependent quantities by taking relaxation time to be constant. For suitable thermoelectric materials, low resistivity and thermal conductivity values are required, along with high Seebeck coefficient values (S) values. The calculated spectra of thermoelectric parameters for La_2_Tm_2_O_7_ (Tm = Hf, Zr) are presented in Figure 8a–e.

The ability of the material to conduct electricity can be explained using electrical conductivity σ. The calculated spectra of electrical conductivity σ for La_2_Tm_2_O_7_ (Tm = Hf, Zr) are presented in Figure 8a. A temperature range of 0 to 1000 K is used to plot calculated thermoelectric parameters. In the spectra of electrical conductivity σ, it is evident that initially, values of σ are zero, and peaks emerge from 650 and 600 K for La_2_Hf_2_O_7_ and La_2_Zr_2_O_7_, respectively. From Figure 8a, we can note an exponential increase in the spectra of σ with increasing temperature values. The semiconductor nature of the pyrochlore oxides La_2_Tm_2_O_7_ (Tm = Hf, Zr) is evident from the increasing trend in the spectra of σ. In the entire temperature range, it is evident from the spectra that values of σ for La_2_Hf_2_O_7_ are less than that of La_2_Zrf_2_O_7_. The maximum values of σ are $4.34\times 10^{16}$ and $6.53\times 10^{16}$ ${\left(Ω.m.s\right)}^{-1}$ for La_2_Hf_2_O_7_ and La_2_Zr_2_O_7_, respectively.

For semiconductors, thermal conductivity (κ) comes from electrons and lattice vibrations. The thermal conductivity (κ) can be calculated using Fourier law. The calculated spectra of electronic thermal conductivity $\kappa_{e}$ for La_2_Tm_2_O_7_ (Tm = Hf, Zr) are presented in Figure 8b. In the spectra of thermal conductivity $\kappa_{e}$, it is evident that initially, values of $\kappa_{e}$ are zero, and then, peaks start to emerge from 600 and 550 K for La_2_Hf_2_O_7_ and La_2_Zr_2_O_7_, respectively. From Figure 8a, we can note an exponential increase in the spectra of $\kappa_{e}$ with increasing temperature values. At room temperature, the phonon wavelength is more significant than the crystal boundary, which results in an increasing trend in the spectra of $\kappa_{e}$. The crystal boundary is greater than or equal to all contributions made by the phonons to $\kappa_{e}$. Temperature and phonon wavelength are inversely proportional to each other. Furthermore, in the entire temperature range, it is evident from the spectra that values of $\kappa_{e}$ for La_2_Hf_2_O_7_ are less than that of La_2_Zrf_2_O_7_. The maximum values of $\kappa_{e}$ are $1.94\times 10^{13}$ and $2.56\times 10^{13}$ $\left(\frac{W}{m.K.s}\right)$ for La_2_Hf_2_O_7_ and La_2_Zr_2_O_7_, respectively.

On joining two different materials, due to temperature gradient, electrons from the high-temperature region move toward the low-temperature region, establishing a potential difference (ΔV) [4]. A ratio of ΔV to ΔT, known as the Seebeck coefficient $\left(S=\frac{\Delta V}{\Delta T}\right)$ can be used to determine the effectiveness of the thermocouple. The calculated spectra of the Seebeck coefficient (S) for La_2_Tm_2_O_7_ (Tm = Hf, Zr) are presented in Figure 8c. The p-type nature of the studied pyrochlore oxides is evident from the positive values of the Seebeck coefficient (S). With the rising temperature, the spectra of S exhibit an exponential decline, a feature of semiconductor devices. The maximum values of S are $2.75\times 10^{-3}$ (200 K) and $2.68\times 10^{-3}$ (200 K) $\left(\frac{V}{K}\right)$ for La_2_Hf_2_O_7_ and La_2_Zr_2_O_7_, respectively. Materials with S values greater than 200 $\frac{\mu V}{K}$ are considered suitable thermoelectric materials. The values of S for both pyrochlore oxides are close to the defined standard. Therefore, these materials can be considered potential thermoelectric materials for device applications. Thermoelectric parameters are also plotted against the chemical potential at 300 K. We can note peaks in both positive and negative regions of Figure 9a; therefore, these materials can be tuned as p-type or n-type depending on our requirement.

The calculated spectra of power factor PF for La_2_Tm_2_O_7_ (Tm = Hf, Zr) are presented in Figure 8d. In the spectra of power factor PF, it is evident that initially, values of PF are zero, and then, peaks start to emerge from 550 and 500 K for La_2_Hf_2_O_7_ and La_2_Zr_2_O_7_, respectively. From Figure 8d, we can note an exponential increase in the spectra of PF with increasing values of temperature. The maximum values of PF are $1.92\times 10^{10}$ and $2.52\times 10^{10}$ $\left(\frac{W}{m.K^{2}.s}\right)$ for La_2_Hf_2_O_7_ and La_2_Zr_2_O_7_, respectively.

Finally, the calculated spectra of the figure of merit ZT for La_2_Tm_2_O_7_ (Tm = Hf, Zr) are presented in Figure 8e The device’s performance can be characterized by a thermoelectric quantity known as the figure of merit ZT. The ZT values start high and then gradually decrease as the temperature rises. The maximum value of ZT is 0.997 (200 K) for pyrochlore oxides La_2_Tm_2_O_7_ (Tm = Hf, Zr). The materials employed in thermoelectric devices must have a ZT value of at least 1 or greater than 1. Pyrochlore oxides La_2_Tm_2_O_7_ (Tm = Hf, Zr) have a ZT value of approximately 1. Hence, these compounds are potential candidates for thermoelectric devices. From the ZT spectra, La_2_Zrf_2_O_7_ is the most promising candidate for thermoelectric device applications. We can predict that these materials are efficient thermoelectric materials by analyzing the figure of merit ZT spectra presented in Figure 9c.

### 2.5. Thermodynamic Properties

The thermodynamic stability of the pyrochlore oxides La_2_Tm_2_O_7_ (Tm = Hf, Zr) was determined by investigating specific heat ($C_{V}$), thermal expansion, and Gibbs free energy (G). The Gibbs2 code is used to determine thermodynamic parameters using quasi-harmonic Debye approximation [30]. Information regarding the required heat energy for a specific material to raise its temperature by 1 K by keeping volume constant is known as specific heat capacity at constant volume ($C_{V}$). The calculated spectra of $C_{V}$ as a function of pressure for La_2_Tm_2_O_7_ (Tm = Hf, Zr) are presented in Figure 10 on a continuous temperature range. Initially, an exponential increase in the values of $C_{V}$ is evident from Figure 10. At higher temperatures, the values of $C_{V}$ become constant with increasing temperatures. The classical behavior of both compounds is evident from the $C_{V}$: $T^{3}$ and Dulong–Petit law spectra at high and low temperatures, respectively [31].

The information regarding the maximum amount of work that a closed thermodynamical system may perform at constant pressure and temperature can be obtained from the thermodynamical quantity known as Gibbs free energy (G). The calculated spectra of G as a function of pressure for La_2_Tm_2_O_7_ (Tm = Hf, Zr) are presented in Figure 11 on a continuous temperature range. An indirect and direct relation of Gibbs free energy is evident from Figure 11 with temperature and applied temperature. At constant pressure and temperature, the value of reversible work that a system can perform can also be obtained from the spectra that a system can perform G.

The information regarding the variations in the volume and shape of the material at constant pressure with changing temperature can be obtained from the thermodynamical parameter known as thermal expansion coefficient (α). The calculated spectra of α as a function of pressure for La_2_Tm_2_O_7_ (Tm = Hf, Zr) are presented in Figure 12 on a continuous temperature range. An exponential increase in the values of α is evident from Figure 12 and shows that La_2_Tm_2_O_7_ (Tm = Hf, Zr) absorbs maximum heat in this region. The values of α and applied pressure are inversely proportional to each other. From Figure 12, we can note that these materials absorb constant heat at higher temperatures, as the curves are parallel to the temperature axis. We can conclude that La_2_Tm_2_O_7_ (Tm = Hf, Zr) are thermodynamically stable compounds based on these thermodynamical parameters.

## 3. Materials and Methods

The full potential linearized augmented plane wave (FP-LAPW) method [32] is used within the framework of density functional theory (DFT) [33,34] to minimize the forces acting on the atoms of the crystal. Using the relaxed geometry, the ground state properties such as structural, optoelectronic, and thermoelectric use first-principles-based WIEN2K code [35]. To analyze the physical properties of La_2_Tm_2_O_7_ (Tm = Hf, Zr), the exchange and correlation potential are calculated by employing Hubbard correction (U)-added generalized gradient approximation (GGA) [24]. Generally, band gap values are underestimated by simple GGA approximation. To overcome this issue, we added a U correction to correct the estimated values of energy bandgaps. While employing the FP-LAPW method, the electrons in the cluster are clustered as valance (electrons in the interstitial region) and core (electrons in muffin-tin spheres) electrons. The plane wave basis set shown in Equation (13) is used to expand the wave function in the interstitial region.
(13)$$V\left(r\right)=\sum_{k}V_{k}e^{i\vec{k}.\vec{r}}$$

On the other hand, the product of spherical harmonics ($Y_{lm})$ and the radial solution for the Schrodinger wave equation ($V_{lm}$) shown in Equation (14) is used to define potential inside the MT spheres.
(14)$$V\left(r\right)=\sum_{l,m}V_{lm}\left(r\right)Y_{lm}\left(r\right)$$

The value of $R_{MT}\times K_{max}=7.0$ is set as the cut-off criteria for the convergence of energy eigenvalues. $K_{max}$ and $R_{MT}$ represent the largest Fermi vector and smallest muffin-tin radii, respectively. A dense k-mesh of 500 k-points is used in these calculations. For self-consistent field calculations, the convergence criterion is less than $10^{-3}$ Ry. The effective Hubbard potential is the difference between Coulomb interaction (U) and exchange potential (J), i.e., $U_{eff}=U-J$. In this study, the value of effective U is set as 7.0 eV by taking U=7.0 eV and J=0 eV. Equation (15) can obtain total energy while using GGA + U formulism.
(15)$$E=E_{0}+E_{GGA+U}$$
where
(16)$$E_{GGA+U}=\frac{U-J}{2}\left(N-\sum^{\;}n_{m,\sigma }^{2}\right)$$

The electrical conductivity, Seebeck coefficient, thermal conductivity, power factor, and figure of merit are generally used to evaluate the thermoelectric performance of crystalline materials. Botlztrap code is employed to calculate thermoelectric parameters by combining DFT with Boltzmann transport theory [28]. The crystalline unit cell structures of pyrochlore oxides La_2_Tm_2_O_7_ (Tm = Hf, Zr) are presented in Figure 13.

## 4. Conclusions

This manuscript investigates pyrochlore oxides’ structural, optoelectronic, and thermoelectric properties using first-principles-based DFT calculations. This study reveals that the compounds mentioned above are promising candidates for photovoltaic and thermoelectric device applications. Analyzing the investigated structural properties, we have concluded, based on ground state energy, that La_2_Hf_2_O_7_ is more stable than La_2_Zr_2_O_7_. Energy band structures exhibit that La_2_Hf_2_O_7_ and La_2_Zr_2_O_7_ are direct bandgap materials with 4.45 and 4.40 eV, respectively. It was established by exploring TDOS spectra that these pyrochlore oxides are nonmagnetic compounds. Based on $\varepsilon_{2}\left(\omega \right)$, we can report that these pyrochlore oxides can efficiently absorb photons in the UV energy range. These materials are ranked as active optical materials, as their refractive index values are between 1.0 and 2.0. The values of n(ω) are 1.5 and 1.52 for La_2_Hf_2_O_7_ and La_2_Zr_2_O_7_, respectively. It is evident from the R(ω) spectra that these compounds are suitable for photovoltaic device applications, as their reflectivity value is negligible (∼20%) in the entire energy range. Thermoelectric properties of La_2_Tm_2_O_7_ (Tm = Hf, Zr) are calculated and discussed using the Boltzmann transport theory employed in the Boltztrap code. The Seebeck coefficient (S) positive values confirm the p-type nature of these semiconductors. The S values are in the range of defined standards for both compounds, confirming their potential to be used in thermoelectric devices. The calculated values of ZT are also around one, which means their conversion efficiency is also good.

## Figures and Tables

**Figure 1 ijms-23-15266-f001:**
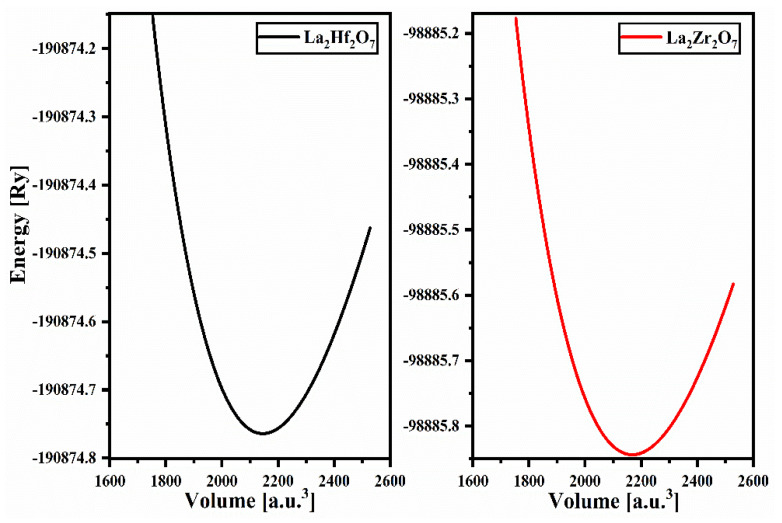
Optimized volume versus energy curves for pyrochlore oxides La_2_Tm_2_O_7_ (Tm = Hf, Zr).

**Figure 2 ijms-23-15266-f002:**
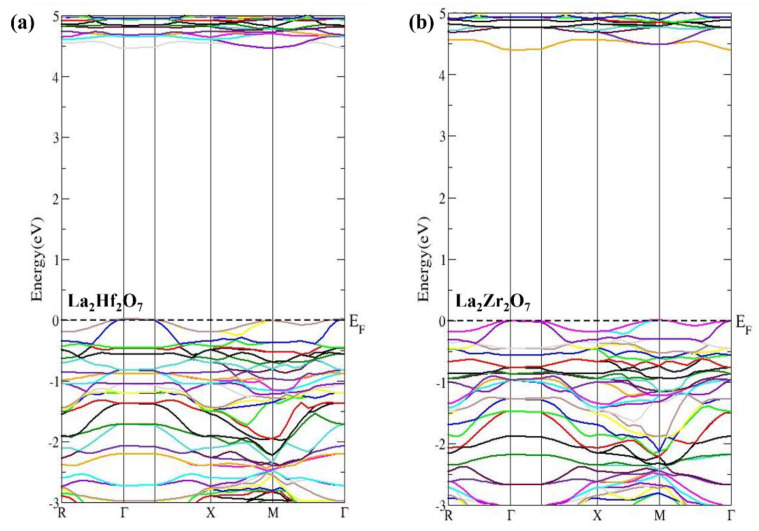
Calculated energy band dispersions for pyrochlore oxides (**a**) La_2_Hf_2_O_7_ and (**b**) La_2_Zr_2_O_7_.

**Figure 3 ijms-23-15266-f003:**
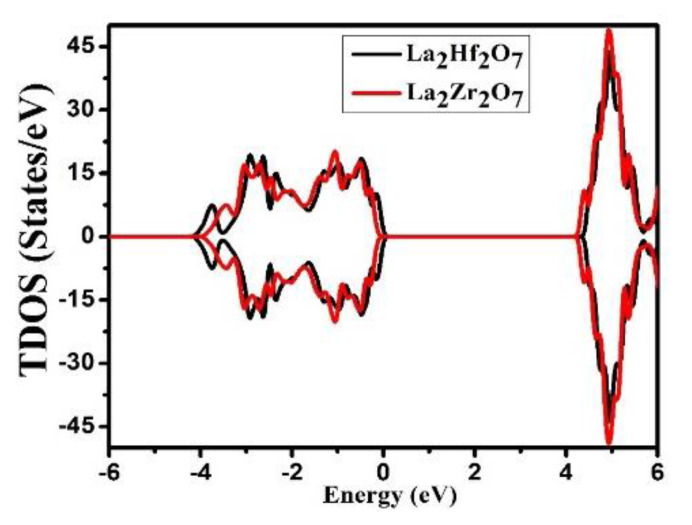
Calculated spectra of the total density of states (TDOS) for pyrochlore oxide La_2_Tm_2_O_7_ (Tm = Hf, Zr).

**Figure 4 ijms-23-15266-f004:**
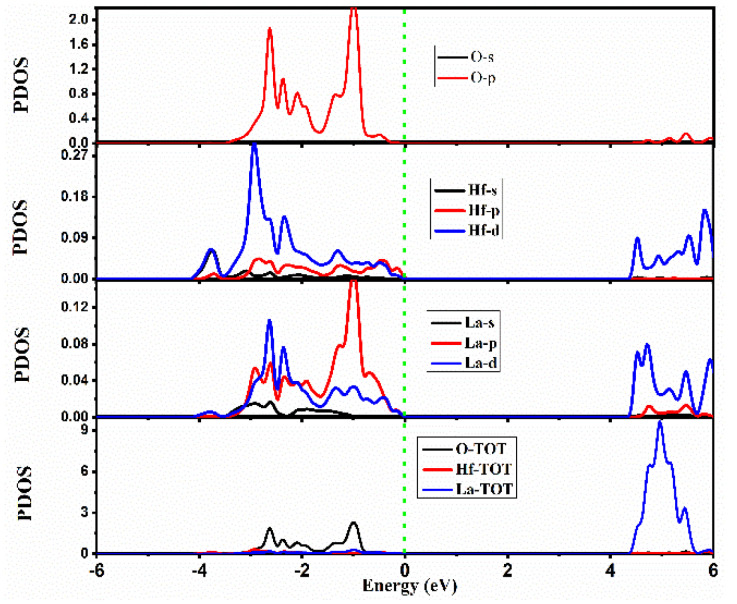
Calculated spectra of the partial density of states (PDOS) for La_2_Hf_2_O_7_.

**Figure 5 ijms-23-15266-f005:**
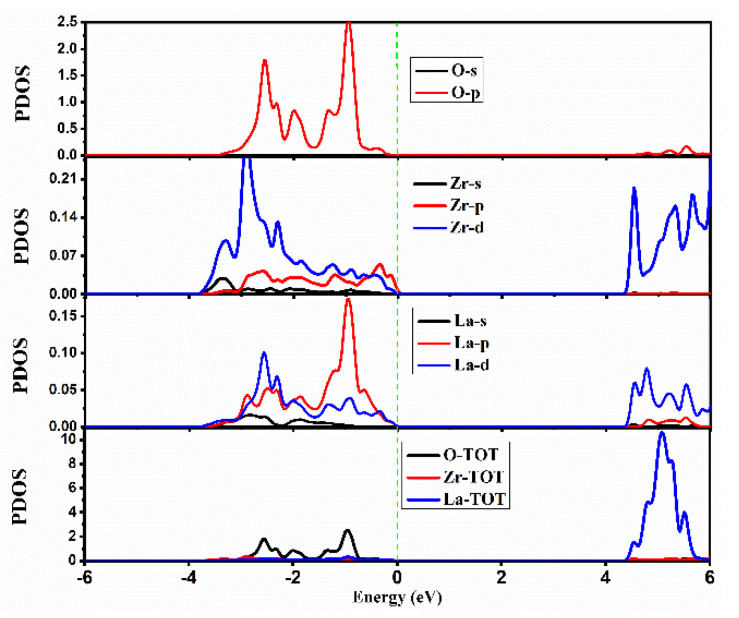
Calculated spectra of the partial density of states (PDOS) for La_2_Zr_2_O_7_.

**Figure 6 ijms-23-15266-f006:**
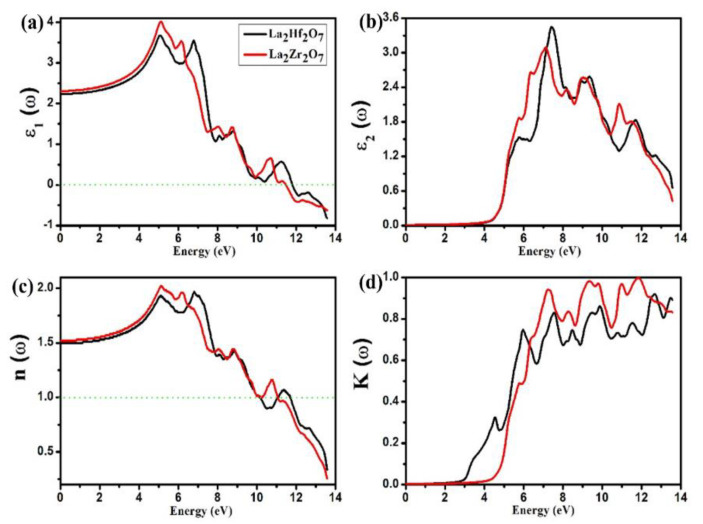
Calculated spectra of (**a**) $\varepsilon_{1}\left(\omega \right)$, (**b**) $\varepsilon_{2}\left(\omega \right)$, (**c**) n(ω) and (**d**) K(ω) for La_2_Tm_2_O_7_ (Tm = Hf, Zr).

**Figure 7 ijms-23-15266-f007:**
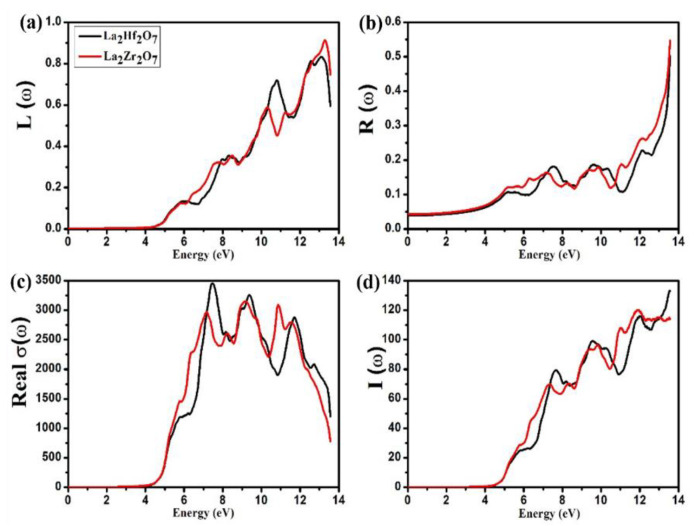
Calculated spectra of (**a**) R(ω), (**b**) L(ω), (**c**) σ(ω), and (**d**) I(ω) for La_2_Tm_2_O_7_ (Tm = Hf, Zr).

**Figure 8 ijms-23-15266-f008:**
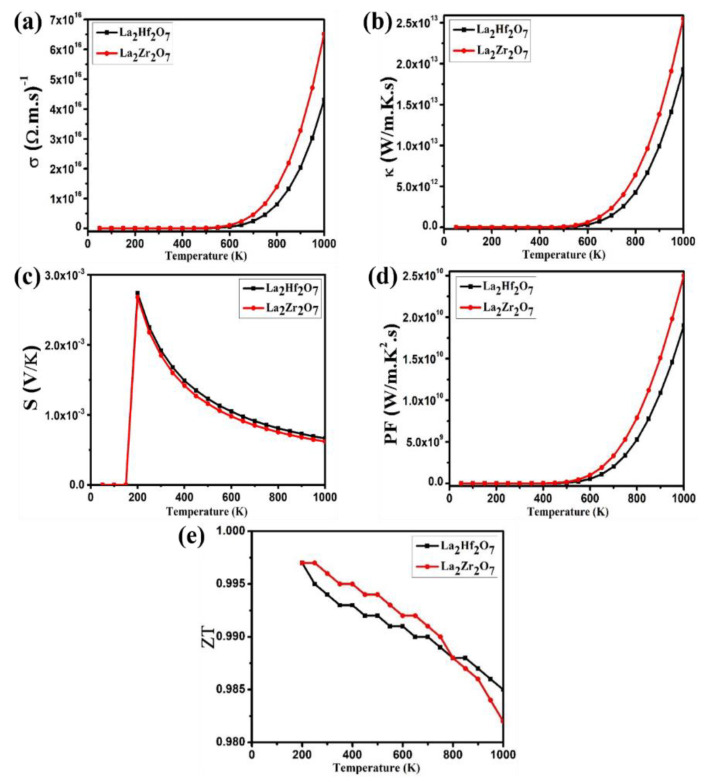
Calculated spectra of (**a**) electrical conductivity (σ), (**b**) electronic thermal conductivity $\left(\kappa_{e}\right)$, (**c**) Seebeck coefficient (S), (**d**) power factor (PF), and (**e**) figure of merit (ZT) for La_2_Tm_2_O_7_ (Tm = Hf, Zr).

**Figure 9 ijms-23-15266-f009:**
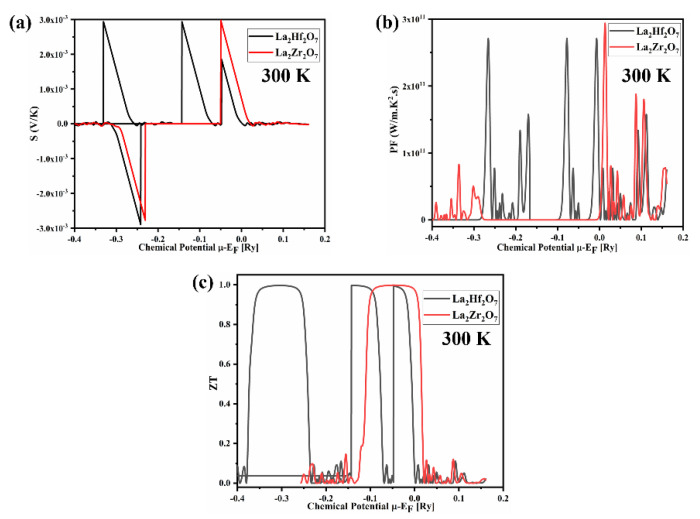
Calculated spectra of (**a**) Seebeck coefficient (S), (**b**) power factor (PF), and (**c**) figure of merit (ZT) as a function of the chemical potential for La_2_Tm_2_O_7_ (Tm = Hf, Zr).

**Figure 10 ijms-23-15266-f010:**
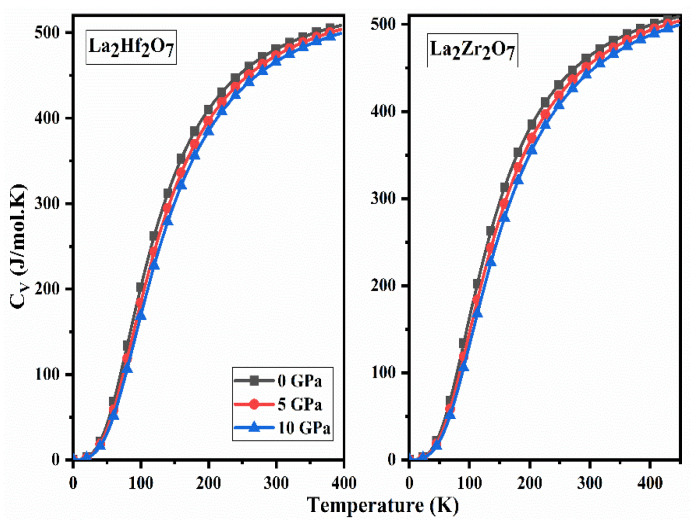
Calculated spectra of specific heat capacity at constant volume $\left(C_{V}\right)$ for La_2_Tm_2_O_7_ (Tm = Hf, Zr).

**Figure 11 ijms-23-15266-f011:**
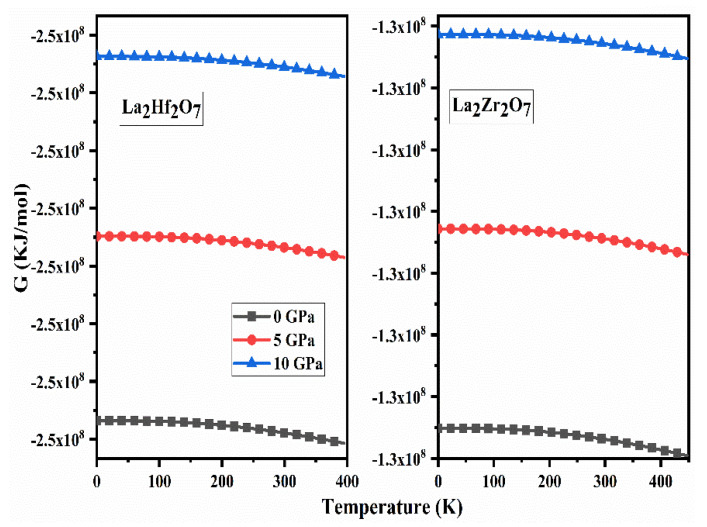
Calculated spectra of Gibbs free energy (G) for La_2_Tm_2_O_7_ (Tm = Hf, Zr).

**Figure 12 ijms-23-15266-f012:**
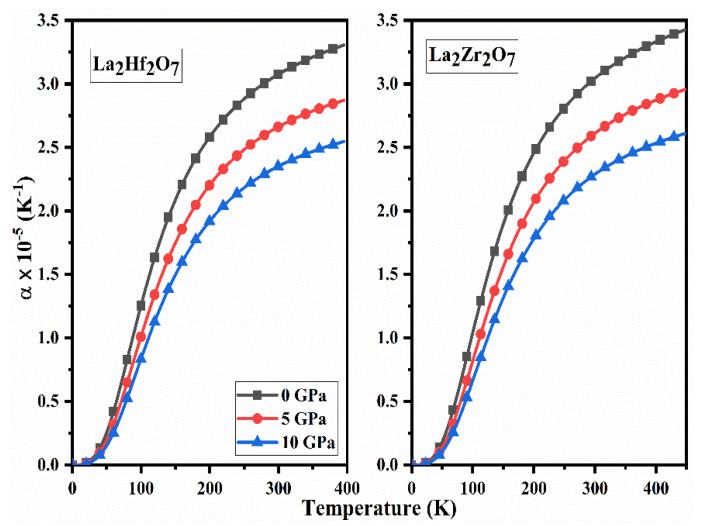
Calculated spectra of thermal expansion (α) for La_2_Tm_2_O_7_ (Tm = Hf, Zr).

**Figure 13 ijms-23-15266-f013:**
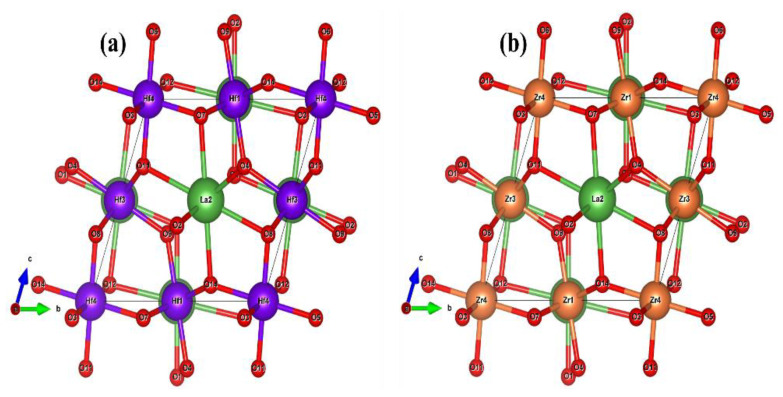
Unit cell structures for crystalline (**a**) La_2_Hf_2_O_7_ and (**b**) La_2_Zr_2_O_7_.

**Table 1 ijms-23-15266-t001:** Optimized parameters for pyrochlore oxides La_2_Tm_2_O_7_ (Tm = Hf, Zr).

	a (Å)	$V_{0}$ (Å)^3^	*B* (GPa)	*B*′ (Gpa)	$E_{0}(\mathrm{Ry})$
La_2_Hf_2_O_7_	6.83	2145.42	173.39	4.38	−190,874.76
La_2_Zr_2_O_7_	6.85	2167.93	167.77	4.39	−98,885.84

**Table 2 ijms-23-15266-t002:** Calculated geometric positions (Å) of La_2_Tm_2_O_7_ (Tm = Hf, Zr).

Element	X	Y	Z
O1	0.625	0.625	0.625
O2	0.375	0.375	0.375
O3	0.332	0.918	0.918
O4	0.082	0.668	0.668
O5	0.918	0.332	0.918
O6	0.918	0.918	0.332
O7	0.332	0.332	0.918
O8	0.332	0.918	0.332
O9	0.918	0.332	0.332
O10	0.668	0.668	0.082
O11	0.082	0.082	0.668
O12	0.668	0.082	0.082
O13	0.668	0.082	0.668
O14	0.082	0.668	0.082
Hf1/Zr1	0.000	0.500	0.000
Hf2/Zr2	0.500	0.000	0.000
Hf3/Zr3	0.000	0.000	0.500
Hf4/Zr4	0.000	0.000	0.000
La1	0.500	0.000	0.500
La2	0.000	0.500	0.500
La3	0.500	0.500	0.000
La4	0.500	0.500	0.500

## Data Availability

Not applicable.

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
