# Peer review of "A First-Principles Investigation on the Structural, Optoelectronic, and Thermoelectric Properties of Pyrochlore Oxides (La2Tm2O7 (Tm = Hf, Zr)) for Energy Applications"

_ijms, 2022, doi:10.3390/ijms232315266_

Round 1

Reviewer 1 Report

In the manuscript, the authors did the detailed first principle study of Pyrochlore oxides La2Tm2O7 (Tm = Hf, Zr). The structural, optoelectronic, and thermoelectric properties of both oxides are calculated and compared. The results of this manuscript are acceptable for publication.

Although the author did a detailed study the relevant pieces of literature references of the previous studies are not reported in the introduction part or in the result sections during the explanation of the different results. Authors should add the Pyrochlore oxides La2Tm2O7 (Tm = Hf, Zr) related previous studies and a clear explanation of the aim of the paper that will help a wider audience to cite the article. 

Author Response

Detailed response to the refree comments are addressed in the attached file.

Reviewer 2 Report

In this work, the authors have studied the structural, optoelectronic and thermoelectric characteristics of the newly designed pyrochlore oxides La2Tm2O7 (Tm = Hf, Zr) using the first-principles calculations based DFT investigations.

The results seem to be very interesting and worth to be published in this journal. Nevertheless, there are some points needing to be clarified before the acceptance of this work.

1-             The authors must give more information in the introduction section on the optoelectronic properties of pyrochlore oxides.

2-             The thermodynamic stability is very important for theoretical predicted structure and should be examined for this.

3-             In figure 8b; The authors present the thermal conductivity.

As defining; K is the thermal conductivity of the material, is composed of the electronic thermal conductivity (Ke) due to the electronic carriers and the lattice thermal conductivity (Kl) due to lattice vibrations, that expressed together by the following equation: k= ke + kl. Moreover, based on the Boltztrap code, we can calculate only the ke (electronic thermal conductivity) which mean also the electronic Figure of merit.

The authors must clarify, if they present the total thermal conductivity which mean that is necessary to add the lattice thermal conductivity or change the name of thermal conductivity by the electronic thermal conductivity and the electronic Figure of merit.

4-             For the thermoelectric properties, the authors should present the thermoelectric coefficients as a function of chemical potential at 300 K.

5-             The manuscript should be written properly (Quality of Figures, mathematical expressions, …), and the English language requires improvements.

In summary, this work can be published provided that the authors give satisfying responses.

Author Response

Detailed Response to the Refree report is available in the attached response file.
